# Effects of Gut Bacteria on the Fitness of Rice Leaf Folder *Cnaphalocrocis medinalis*

**DOI:** 10.3390/insects15120947

**Published:** 2024-11-29

**Authors:** Qinjian Pan, Qingpeng Wang, Ikkei Shikano, Fang Liu, Zhichao Yao

**Affiliations:** 1Institutes of Agricultural Science and Technology Development, Joint International Research Laboratory of Agriculture and Agri–Product Safety, Yangzhou University, The Ministry of Education of China, Yangzhou 225009, China; 2College of Plant Protection, Yangzhou University, Yangzhou 225009, China; 3Department of Plant and Environmental Protection Sciences, University of Hawai’i at Mānoa, Honolulu, HI 96822, USA

**Keywords:** rice leaf folder, gut bacteria, microbial community, single isolate, entomopathogenicity, fecundity, development

## Abstract

Most insects harbor diverse bacterial communities in their guts, which can influence host behavior and physiology. The rice leaf folder *C. medinalis*, a significant migratory pest in Asia, also possesses diverse bacterial communities, but the functions of these bacteria in modulating host’s growth and fitness remain unclear. In this study, we isolated 15 bacterial species from field–collected *C*. *medinalis* larvae using a culture–dependent method. When *C. medinalis* larvae were orally inoculated with individual bacterial isolates, we found that many isolates had nonessential or deleterious roles in the development and reproduction of *C. medinalis*. Among these isolates, 10 caused high mortality, highlighting their potential as biocontrol agents. Antibiotic treatment to eliminate gut bacteria negatively affected the fitness of *C. medinalis*, which included prolonged larval and pupal development, and decreased adult longevity and fecundity. Inoculation of a bacterial community to antibiotic–treated larvae recovered some of the negative effects, including fecundity. Our results reveal that *C. medinalis* possesses bacteria capable of both mutualistic and pathogenic interactions.

## 1. Introduction

The research over the past two decades has revealed functional relationships between some of the gut microorganisms and their insect hosts [1,2,3]. Some bacterial isolates can directly influence the fitness of their host insects by aiding protein digestion, fixing nitrogen, and producing essential nutrients [4,5,6,7]. Other isolates may indirectly influence the growth of insect populations by detoxifying toxic plant chemicals, degrading insecticides, and boosting the immune systems of host insects [8,9,10]. Since insects harbor an astonishing diversity of microbes in their guts, the microbial community may contribute to maintaining host homeostasis and health by regulating nutrient metabolism and immune system responses [11,12,13]. Hence, enhancing our understanding of the roles of individual bacteria and bacterial communities in the growth and reproduction of their insect hosts may be helpful in developing methods to regulate insect pest populations.

Although gut microbes are usually commensal in healthy herbivores, their roles can change from beneficial to harmful depending on host condition. Some gut bacteria may emerge as opportunistic pathogens if the host’s modulation of intestinal microbes deteriorates [14,15]. For example, the effects of *Enterococcus* and *Serratia* isolates collected from the guts of healthy *Manduca sexta* and *Spodoptera frugiperda* larvae changed from commensal to pathogenic when the larvae were fed a toxic diet [16,17]. Thus, a better understanding of the functions of gut bacteria for host survival would aid in developing pathogenic bacteria as biocontrol agents. Recent advancements in next–generation sequencing (NGS) technology have revealed a diversity of microbial communities associated with insects [18], yet the research on the functions of these gut microbes remains limited. The culture–dependent method is another option for identifying bacteria, but it has limitations due to culturing conditions, such as temperature, nutrition, and humidity, which may prevent the growth and identification of many bacteria species. Despite this major drawback, a huge advantage of this method is that it offers the possibility to culture specific bacteria and allows for assessments of bacterial functions. In this study, we investigated the positive and negative effects of gut bacterial isolates and community on the fitness and reproduction of a major lepidopteran rice pest.

*Cnaphalocrocis medinalis* (Lepidoptera: Pyralidae) is a devastating pest of rice (*Oryza sativa*), affecting yields significantly across Asia [19]. The larvae can cause extensive damage to rice leaves by rolling and defoliating them, ultimately leading to significant yield losses, reduced crop quality, increased vulnerability to diseases, and additional economic burdens on farmers [20,21]. Moreover, *C. medinalis* adults have a strong migratory propensity and flight capability [22]. To our knowledge, there are few effective commercial microbial insecticides and biocontrol agents to control this pest. Farmers often resort to chemical insecticides, but their excessive use has led to adverse effects on the environment and human health. Therefore, there is a pressing need to develop alternative, safe, sustainable, and eco–friendly strategies to manage *C. medinalis* infestations.

Existing reports on gut microbes of *C. medinalis* have primarily used NGS technology to compare variations in gut bacteria species diversity associated with differences in host plants, geographic locations, and life stages [23,24,25]. However, the role of gut bacteria in modulating the fitness of *C. medinalis* has not been explored explicitly. In the present study, we firstly employed both culture–dependent method and 16S rRNA sequencing to identify culturable bacterial isolates from the guts of field–collected *C. medinalis* larvae. Secondly, we tested the effects of each bacterial isolate on the fitness of *C. medinalis* larvae and adults. Thirdly, we evaluated the function of a bacteria community on the development and reproduction of *C. medinalis*. We aimed to understand the intricate interplay between *C. medinalis* and its gut bacteria by investigating the importance of gut bacteria on host fitness, providing insights into potential pest management strategies, such as identifying potential microbial control agents or eliminating essential bacteria from the gut using antibiotics.

## 2. Materials and Methods

### 2.1. Insects

*Laboratory colony of C. medinalis used for bacteria inoculation.* Larvae and pupae of wild *C. medinalis* were initially collected from a paddy field in Yangzhou University, China, in September 2017. Larvae were reared on wheat (*Triticum aestivum* cv. Zhenmai 168) seedlings for more than 30 generations under the laboratory condition of 27 ± 1 °C, 75% RH, and a photoperiod of 14:10 h (light:dark). To obtain eggs, groups of fifteen mating pairs of newly emerged moths (1:1) were placed in plastic cups (10.8 cm in diameter, 14.2 cm in height), which were covered with a transparent plastic film. Adults were provided with 10% honey solution as supplementary nutrition. *C. medinalis* larvae underwent five larval instars under laboratory conditions.

*Field*–*collected C. medinalis larvae used for culturing gut bacterial isolates.* Wild fourth instar *C. medinalis* larvae (n = 45) were collected from a paddy field in Jiangyan (120.10° E, 32.57° N), Taizhou, China, in July 2020. They were stored in 50 mL centrifuge tubes with ventilation holes, punctured using a stainless–steel insect pin (size 3).

### 2.2. Gut Dissection and Bacteria Sampling

Field–collected larvae were starved for 4 h, sterilized externally with 75% ethanol, and rinsed with sterilized water three times. The entire gut tissue from each larva was dissected using two forceps by gently holding both ends of the larva and pulling in opposite directions, then transferred into a sterile 1.5 mL centrifuge tube containing 0.5 mL of sterile PBS buffer (pH 7.4). Gut tissue was ground manually with a sterile pestle and maintained at 37 °C for 1 h to release bacteria. The homogenization buffer was centrifuged at 5000× *g* for 30 min to remove debris (i.e., frass and gut tissue). The supernatants from the homogenization buffers of each larval gut tissue were pooled together (~22.5 mL) and transferred into a sterile 50 mL centrifuge tube. We used 20 mL of the bacterial suspension in the following experiments: 5 mL was used to identify culturable bacterial isolations, while the remaining 15 mL served as a source of bacterial community to test their effects on *C. medinalis* fitness.

### 2.3. Bacteria Isolation and Identification

The bacterial isolates were cultured and identified according to Pan et al. [26] with minor modifications. Briefly, tenfold serial dilutions of the supernatant containing gut bacteria were prepared and 5–6 dilutions were separately cultured on the surface of 2 × YT agar plates (containing 1.6% tryptone, 1.0% yeast extract, 0.5% NaCl, and 1.4% agar). All cultures were incubated at 27 °C for roughly 12 h. Different bacterial isolates were chosen depending on the morphological characteristics (color, shape, and size). Selected isolates were purified by sub–culturing twice. The purified isolates were individually grown in liquid 2 × YT media (containing 1.6% tryptone, 1.0% yeast extract, and 0.5% NaCl) in a rotary shaker at 200 rpm at 27 °C overnight. Twenty microliters of each liquid bacterial isolate were transferred into a sterile PCR tube, heated at 95 °C for 10 min to release the DNA from bacterial cells. The bacterial DNA was amplified with a PCR reaction using universal 16S rRNA primers (27F 5′–AGAGTTTGATCCTGGCTCAG–3′, 1492R 5′–GGTTACCTTGTTACGACTT–3′). The PCR products were sequenced at Beijing Tsingke Biotech Co., Ltd. (Nanjing, China). The obtained 16S rRNA sequences were analyzed with NCBI (National Center for Biotechnology Information) and RDP (Ribosomal Database Project). All sequences of the bacterial isolates were submitted to the GenBank database under the accession numbers PP955231–PP955245, as shown in Table 1. Each bacterial isolate (500 μL) was stored in 500 μL 25% sterile glycerol at −80 °C for future use.

### 2.4. Effects of Each Bacterial Isolate on C. medinalis Fitness

A total of 15 bacterial isolates were identified. *C. medinalis* larvae were inoculated with individual bacterial isolates to determine the effects of a single isolate on subsequent developmental stages. The 15 isolates were separated into 3 groups, such that 5 isolates and a control were tested at each time.

The bacterial suspensions for larval inoculation were prepared by growing stock bacteria in liquid 2 × YT media overnight to obtain an optical density (OD) of 1 at a wavelength of 600 nm, which produces approximately 1 × 10^9^ colony forming units per milliliter [26]. Fresh wheat leaves (~10 cm in length) were surface sterilized by submerging in 75% ethanol and then rinsing with sterile distilled water three times. The sterile leaves were dipped into 50 mL of one bacterial isolate suspension (treated leaves) or distilled water (untreated leaves) for 5 min, allowed to air–dry for 30 min at room temperature, and transferred into sterile Petri dishes (10 cm in diameter). The petioles of these treated or untreated leaves were inserted into a moist, sterile, cotton ball to keep fresh.

Eggs from the *C. medinalis* lab colony were surface sterilized by 2% bleach. One neonate that newly hatched from surface sterilized eggs was introduced into a sterile Petri dish with treated or untreated leaves. A total of 30 newly hatched larvae were used in each treated and untreated group. The treated or untreated leaves were replaced with corresponding leaves of the same treatment every two days until pupation or death. All larvae either pupated or died within 20 days. The resulting pupae were paired to allow one male and one female from a treatment to mate and lay eggs in a 320 mL plastic cup. Each cup was covered with plastic film containing a cotton ball that was pre–soaked with 10% honey solution. Each insect was monitored daily until all individuals had died. Detailed fitness parameters were investigated as follows: the development time of each larval and pupal stage, pupal weight and sex, and adult longevity; pre–oviposition and oviposition period; and fecundity of each female (total number of eggs laid until death). To avoid bias from individuals that died before reaching the pupal stage, data from those that did not survive pupation in the treated and untreated groups were excluded from analyses of developmental measures. The numbers of eggs deposited in each cup were counted daily until the female died. Females that did not lay eggs were excluded from fecundity analyses. Pupae were weighed using a micro–balance (resolution 0.1 mg; AL204–IC, Mettler Toledo Technology Co., LTD (Shanghai, China). The sex of each pupa was identified according to Chen et al. [27] and the proportion of adult females was determined for each treatment.

### 2.5. Effects of Gut Bacterial Community on C. medinalis Fitness

To examine the effects of bacterial community on the fitness of *C. medinalis* larvae, we applied an antibiotic (AB) cocktail to reduce the gut bacterial community. Three antibiotics, neomycin trisulfate salt hydrate (A610366), chlortetracycline hydrochloride (A600297), and streptomycin sulfate (A610494), were all produced by Sangon Biotech. Each milliliter of AB cocktail contained 0.2 mg neomycin trisulfate salt hydrate, 1 mg chlortetracycline hydrochloride, and 0.06 mg streptomycin sulfate dissolved in sterile distilled water. Fresh wheat leaves (~10 cm in length) were surface sterilized in 75% ethanol as described above and dipped into 50 mL of the AB cocktail for 5 min. After ensuring the even spread of the AB cocktail on the leaf surface, the leaves were air–dried at room temperature for 1 h. In order to investigate the effects of antibiotic treatment on the fitness of 4th and 5th instar larvae, a newly molted 4th or 5th instar larva from the laboratory colony was placed in a Petri dish (10 cm in diameter) and received an AB–treated (AB^+^) or untreated (AB^−^) ethanol–sterilized leaf. The AB^+^ and AB^−^ leaves were replaced every two days until the 4th instar larvae initiated their molt into the 5th instar or until the 5th instar larvae initiated pupation. Larvae (n = 30) treated with AB in the 4th instar (AB^+^) were fed leaves surface–sterilized with 75% ethanol during their 5th instar until pupation. Larvae (n = 30) treated with AB in the 5th instar (AB^+^) were fed AB^+^ leaves during their 5th instar until pupation. Therefore, the AB treatment was only performed during one larval instar, either in the 4th instar or 5th instar. AB^−^ larvae (n = 30) were fed ethanol–sterilized leaves until pupation. Parameters of development time, adult longevity, and female fecundity were measured as described above.

To determine how inoculation with a community of gut bacteria affects the growth and reproduction of *C. medinalis*, we transferred the bacterial community obtained from field–collected larvae into lab–reared 5th instar larvae that had been treated with AB in the 4th instar. The bacterial community suspension (see Materials and Methods Section 2.2) was equally divided into thirty 0.5 mL aliquots. Each 0.5 mL aliquot of the bacterial suspension was evenly applied on the upper surface of two ethanol–sterilized leaves (each ~10 cm in length) and air–dried at room temperature for 1 h.

Thirty newly molted 4th instar larvae were placed in individual Petri dishes and fed AB^+^ leaves for the duration of the 4th instar. Once the larvae molted to the 5th instar, they were transferred to new individual Petri dishes and provided 2 leaves treated with the bacterial community suspension. The bacteria–treated leaves were replaced every two days until the larvae pupated. AB^−^ (n = 30) larvae and AB^+^ (n = 30) larvae that were fed ethanol–sterilized leaves (not treated with bacteria) in the 5th instar served as controls. The fitness parameters were measured as described above.

### 2.6. Statistical Analyses

The effects of single bacterial isolates and antibiotic treatments on the duration of different life stages, pupal weight, and fecundity were analyzed with Student’s *t* test to compare each treatment against the control at *α* = 0.05. Survival curves for *C. medinalis* inoculated with individual bacterial isolates were analyzed by Kaplan–Meier survivorship analyses followed by log–rank pairwise comparisons tests. The effects of inoculation with a community of bacteria on the development time of different life stages, pupal weight, and fecundity were analyzed using a one–way analysis of variance (ANOVA). Duncan multiple range tests were used for multiple comparisons at *α* = 0.05. Statistical analyses were performed using IBM SPSS statistical software (version 26).

## 3. Results

### 3.1. Isolation and Identification of Cultured Bacteria from C. medinalis Gut

Fifteen bacterial isolates were identified and listed in Table 1. They were categorized into 3 bacterial phyla, 4 classes, 5 orders, 10 families, and 12 genera (Table 2). The three bacterial phyla were predominant in the gut bacterial community of *C. medinalis*. These isolates belonged to 12 genera consisting of *Acinetobacter* (one isolate), *Klebsiella* (two isolates), *Enterobacter* (two isolate), *Herbaspirillum* (one isolate), *Pseudomonas* (one isolate), *Pantoea* (two isolates), *Stenotrophomonas* (one isolate), *Bacillus* (one isolate), *Chryseobacterium* (one isolate), *Enterococcus* (one isolate), *Providencia* (one isolate), and *Myroides* (1 isolate).

### 3.2. Effects of Single Bacterial Isolate on Development and Reproduction of C. medinalis

#### 3.2.1. Survival of *C. medinalis* During Immature Stage

The survival of *C. medinalis* over a 20 d period was strongly influenced by the ingestion of specific bacterial isolates. *C. medinalis* larvae fed control leaves exhibited a high survival rate, ranging from 96.7% to 93.3%. Feeding larvae with leaves treated with some bacterial isolates did not negatively affect survival. These included *Acinetobacter soli* (16.7% mortality; Figure 1A: *χ*^2^ = 1.357; *p* = 0.244), *E. ludwigii* (6.7% mortality; Figure 1A: *χ*^2^ = 0.001; *p* = 0.9930), *Herbaspirillum huttiense* (16.7% mortality; Figure 1A: *χ*^2^ = 1.366; *p* = 0.242), and *K. pneumoniae* (13.3% mortality; Figure 1B: *χ*^2^ = 0.679; *p* = 0.410). Although there was no statistically significant difference, larval feeding on *K. aerogenes*–treated leaves slightly reduced survival compared to the controls (23.3% mortality; Figure 1A: *χ*^2^ = 3.320; *p* = 0.0680). Ten bacterial isolates significantly decreased survival relative to the control during the 20 d period. These included larvae that fed on leaves treated with *P. mosselii* (56.7% mortality; Figure 1A: *χ*^2^ = 17.404; *p* = 0.0001), *P. dispersa* (40.0% mortality; Figure 1B: *χ*^2^ = 9.483; *p* = 0.0020), *Stenotrophomonas maltophilia* (26.7% mortality; Figure 1B: *χ*^2^ = 4.520; *p* = 0.0340), *Bacillus atrophaeus* (33.3% mortality; Figure 1B: *χ*^2^ = 6.242; *p* = 0.0120), *C. culicis* (40.0% mortality; Figure 1B: *χ*^2^ = 9.956; *p* = 0.0020), *Enterococcus hirae* (23.3% mortality; Figure 1C: *χ*^2^ = 5.169; *p* = 0.0230), *E. asburiae* (20.0% mortality; Figure 1C: *χ*^2^ = 4.434; *p* = 0.0350), *P. ananatis* (46.7% mortality; Figure 1C: *χ*^2^ = 14.801; *p* = 0.0001), *Providencia vermicola* (26.7% mortality; Figure 1C: *χ*^2^ = 6.179; *p* = 0.0130), and *M. odoratus* (40.0% mortality; Figure 1C: *χ*^2^ = 11.512; *p* = 0.0010). *C. medinalis* survival was affected the most by *P. mosselii*, which induced 56.70% mortality.

#### 3.2.2. Development Time of *C. medinalis* Larvae

The duration of the larval stage varied among *C. medinalis* larvae fed leaves treated with different bacterial isolates (Table 3). In group 1, larvae fed leaves treated with *K. aerogenes* spent significantly longer in the fourth instar than the controls (*t*_1, 49_ = −2.855, *p* = 0.0060). Larvae fed leaves treated with *P. mosselii* took significantly longer than the controls in the 3rd (*t*_1, 39_ = −6.086, *p* = 0.0001) and 5th (*t*_1, 39_ = −2.167, *p* = 0.0360) instars. Overall, the total development time of larvae was prolonged when fed leaves treated with *E. ludwigii* (*t*_1, 54_ = −2.295, *p* = 0.0260), *H. huttiense* (*t*_1, 51_ = −2.566, *p* = 0.0130), and *P. mosselii* (*t*_1, 39_ = −4.311, *p* = 0.0001).

In group 2, the duration of the 5th instar was significantly longer in larvae fed leaves treated with *P. dispersa* than control leaves (*t*_1, 44_ = −4.418, *p* = 0.0001). Larvae fed leaves treated with *S. maltophilia* spent longer in the 4th (*t*_1, 48_ = −3.264, *p* = 0.0020) and 5th (*t*_1, 48_ = −2.756, *p* = 0.0080) instars than those fed control leaves. Overall, the total development time of larvae was prolonged if fed leaves treated with *S. maltophilia* (*t*_1, 47_ = −4.671, *p* = 0.0001) and *B. atrophaeus* (*t*_1, 47_ = −2.771, *p* = 0.0080).

In group 3, larvae fed leaves treated with *E. hirae* took significantly longer to complete the 1st (*t*_1, 50_ = −6.026, *p* = 0.0001), 2nd (*t*_1, 50_ = −2.278, *p* = 0.0270), and 5th (*t*_1, 50_ = −2.230, *p* = 0.0300) instars than those fed control leaves. Larvae fed *E. asburiae*–treated leaves took longer to complete the 1st instar than the controls (*t*_1, 46_ = −2.193, *p* = 0.0330), and those fed *P. ananatis*–treated leaves took longer to complete the 1st (*t*_1, 43_ = −2.472, *P* = 0.0420) and 4th (*t*_1, 43_ = −2.586, *p* = 0.0160) instars than the controls. Larvae fed leaves treated with *P. vermicola* had delayed development in the 3rd (*t*_1, 49_ = −3.603, *p* = 0.0010) and 4th (*t*_1, 49_ = −3.035, *p* = 0.0040) instars than the controls, and feeding on *M. odoratus*–treated leaves delayed the development duration in the 5th (*t*_1, 45_ = −2.254, *p* = 0.029) instar relative to the controls. Overall, the complete larval period was significantly prolonged in larvae fed leaves treated with *E. hirae* (*t*_1, 50_ = −5.982, *p* = 0.0001), *P. ananatis* (*t*_1, 43_ = −3.939, *p* = 0.0010), *P. vermicola* (*t*_1, 49_ = −4.694, *p* = 0.0001), and *M. odoratus* (*t*_1, 45_ = −3.254, *p* = 0.0020).

#### 3.2.3. Pupal Time, Pupal Weight, Sex Ratio, Adult Longevity, and Fecundity of *C. medinalis*

Ingestion of bacterial isolates did not significantly affect the duration of the prepupal period (Table 4), but the duration of the pupal stage was significantly prolonged relative to the control in *C. medinalis* fed leaves treated with *E. hirae* (*t*_1, 50_ = −2.670, *p* = 0.0100), *E. asburiae* (*t*_1, 46_ = −2.255, *p* = 0.0290), *P. ananatis* (*t*_1, 43_ = −3.245, *p* = 0.0040), and *P. vermicola* (*t*_1, 49_ = −3.387, *p* = 0.0020). The pupal weight was significantly reduced in *C. medinalis* fed *P. dispersa* (*t*_1, 44_ = −4.480, *p* = 0.0001), *S. maltophilia* (*t*_1, 48_ = 2.359, *p* = 0.0220), *K. pneumoniae* (*t*_1, 52_ = 3.352, *p* = 0.0020), *B. atrophaeus* (*t*_1, 47_ = 3.445, *p* = 0.0010), *C. culicis* (*t*_1, 43_ = 2.652, *p* = 0.0110), and *E. hirae* (*t*_1, 50_ = −2.428, *p* = 0.0190), relative to the control pupae. The proportion of females (i.e., sex ratio) was approximately 0.5 in all treatments except *C. medinalis* fed leaves treated with *B. atrophaeus*, which resulted in a 0.62 female ratio. Some bacterial isolates prolonged the lifespan of the adults. Compared to the controls, adult longevity was longer if, during the larval stage, they were fed leaves treated with *Acinetobacter soli* (*t*_1, 51_ = 3.381, *p* = 0.0010), *K. aerogenes* (*t*_1, 49_ = 2.191, *p* = 0.0330), *H. huttiense* (*t*_1, 51_ = 3.381, *p* = 0.0010), *P. dispersa* (*t*_1, 44_ = 2.964, *p* = 0.0050), and *S. maltophilia* (*t*_1, 48_ = 3.017, *p* = 0.0040). The total numbers of eggs laid (i.e., fecundity) was significantly reduced relative to the controls in females fed *A. soli* (*t*_1, 17_ = 3.156, *p* = 0.0060), *S. maltophilia* (*t*_1, 20_ = 2.706, *p* = 0.0140), *B. atrophaeus* (*t*_1, 23_ = 3.123, *p* = 0.0050), and *P. vermicola* (*t*_1, 21_ = 2.175, *p* = 0.0410). Collectively, some of the bacterial isolates had neutral effects on host fitness, while others adversely affected the development and reproduction of *C. medinalis*.

### 3.3. Effects of Gut Bacterial Community on Development and Reproduction of C. medinalis

#### 3.3.1. Immature Stage

The number of colony–forming units of bacteria in larval guts visibly decreased when *C. medinalis* larvae were fed leaves treated with an antibiotic cocktail (no data; Figure 2). Compared to the controls, when 4th instar larvae were treated with antibiotics, their development time in the 5th instar was significantly prolonged (Figure 3A, *t*_1, 54_ = −6.531, *p* = 0.0001). When larvae were fed antibiotics during the 5th instar, their 5th instar durations were significantly prolonged relative to the controls (Figure 3A: *t*_1, 58_ = 2.576, *p* = 0.0130). When 5th instar larvae that had been treated with antibiotics in the 4th instar were fed leaves treated with the bacterial community, their development was still significantly prolonged relative to the controls (Figure 3B, *F*_2, 83_ = 4.907, *p* = 0.0100).

When larvae were fed antibiotics during either the 4th (Figure 3E, *t*_1, 54_ = −3.375, *p* = 0.0010) or 5th instar (Figure 3E, *t*_1, 58_ = 3.167, *p* = 0.0030), the duration of their pupal stages was significantly extended relative to the controls. Compared to the controls, the pupal weight was significantly reduced when fed antibiotics during the 4th larval instar (Figure 3G *t*_1, 54_ = 2.836, *p* = 0.0060), but was not significantly different from the controls when larvae were fed antibiotics during the 5th instar (Figure 3G, *t*_1, 58_ = −1.251, *p* = 0.2160). Though larvae fed antibiotics during the 4th instar had significantly longer pupal durations (Figure 3F, *F*_2, 83_ = 6.943, *p* = 0.0020) and lighter pupae (Figure 3H, *F*_2, 83_ = 3.176, *p* = 0.0470) compared to the controls, reintroducing the gut bacterial community in the 5th instar recovered these negative effects to some extent, such that the duration of their pupal stage and pupal weight were not significantly different from the controls that were not fed antibiotics (Figure 3F).

#### 3.3.2. Adult Stage

When 4th instar larvae were treated with antibiotics, their adult longevity was significantly shortened relative to the controls (Figure 4A, *t*_1, 54_ = 3.535, *p* = 0.0010). When 5th instar larvae were treated with antibiotics, their adult longevities were not significantly different from the controls (Figure 4A, *t*_1, 58_ = −0.935, *p* = 0.3540). Antibiotic–treated larvae that had their gut bacterial community reintroduced during the 5th instar still exhibited shorter adult longevity relative to the controls (Figure 4B, *F*_2, 83_ = 5.739, *p* = 0.0050).

Fecundity was significantly lower when they were fed antibiotics during the 4th instar (Figure 4C, *t*_1, 44_ = 5.476, *p* = 0.0001), but antibiotic treatment in the 5th instar did not affect fecundity (Figure 4C, *t*_1, 26_ = −0.849, *p* = 0.4030). Interestingly, reintroducing the gut bacterial community to antibiotic–treated larvae during the 5th instar recovered the reproductive capacity of females (Figure 4D, *F*_2, 83_ = 10.874, *p* = 0.0001). The bacterial community–inoculated *C. medinalis* laid 37.6% more eggs than females that were treated with antibiotics in the 4th instar and not given bacteria in the 5th instar. The results indicate that gut bacteria may have a significant influence on the fitness of the insect host during earlier developmental instars. Importantly, the bacterial community may play an important role in the growth of *C. medinalis* populations, suggesting that targeting the gut bacterial community could be an effective strategy to control *C. medinalis* infestation by reducing or eliminating their gut bacteria.

## 4. Discussion

Studies from the last two decades have demonstrated that gut bacteria in insects play important roles in mediating the fitness of their hosts, though their roles in lepidopteran hosts is less clear. In this study, we identified 15 culturable bacterial isolates from the guts of field–collected *C. medinalis* larvae using a culture–dependent method. According to previous publications on the diversity of gut bacteria in *C. medinalis* using NGS technology, six of the bacterial isolates in our study, *K. aerogenes*, *K. pneumoniae*, *E. ludwigii*, *E. asburiae*, *P. dispersa*, and *P. ananatis*, were newly identified [23,24]. *K. aerogenes* has been identified in the guts of *Tuta absoluta* [28]; *K. pneumoniae* and *E. ludwigii* have been identified in the foregut of *Helicoverpa zea* [29,30]; *E. asburiae* has been found in the digestive tract of *Helicoverpa zea*, *Plutella xylostella,* and *Tuta absoluta* [6,28,29]; and *P. dispersa*, and *P. ananatis* have been isolated from larval guts of *S. frugiperda* [31]. The variations in bacterial species among these isolates are significantly influenced by insect species and their living conditions, along with the methods used for identification, whether culture–dependent or –independent.

Our bacterial inoculation experiments revealed that the 15 identified bacterial isolates demonstrated varying degrees of effects on *C. medinalis* fitness. We found that *C. medinalis* females laid significantly fewer eggs if they had been reared on leaves treated with *Acinetobacter soli*, *S. maltophilia*, *B. atrophaeus,* and *P. vermicola* relative to the controls. Though development on leaves treated with *K. aerogenes*, *E. ludwigii*, *H. huttiense,* and *K. pneumoniae* resulted in prolonged larval instars, fecundity was not affected compared to the controls. Importantly, high mortality, ranging from 20.0% to 56.7%, was induced by 10 bacterial isolates: *P. mosselii*, *P. dispersa*, *S. maltophilia*, *B. atrophaeus*, *C. culicis*, *E. hirae*, *E. asburiae*, *P. ananatis*, *P. vermicola*, and *M. odoratus*. Pathogenic or opportunistically pathogenic isolates are common in the guts of lepidopteran larvae [14,16,17]. Some of the bacterial species we isolated are known entomopathogens. *S. maltophilia* is known to induce mortality in insects including cowpea weevil, *Callosobruchus maculatus*, and termites, *Coptotermes heimi* and *Heterotermes indicola* [32,33]. *B. atrophaeus* possesses the capability to generate chitinolytic enzymes, which has been shown to exhibit insecticidal potential [34]. *P. vermicola* has been proven to be pathogenic to many insects, including the greater wax moth, *Galleria mellonella*; beet armyworm, *Spodoptera exigua*; and Mexican fruit fly, *Anastrepha ludens* [35,36]. Hence, the bacterial isolates identified in our study may potentially be developed as biological control agents for managing *C. medinalis*, and possibly other agriculture pests.

Diverse microbial communities in the guts of insect hosts can profoundly influence numerous aspects of host biology [26,37]. In the present study, our colony larvae, cleared of gut bacteria using a cocktail of antibiotics, performed worse than control larvae with intact gut microbiota, despite individual isolates having neutral or negative effects on *C. medinalis* fitness. Additionally, antibiotic treatment in the 4th larval instar had a greater negative effect on adult longevity and fecundity than treatment in the 5th instar. Elimination of gut bacteria is known to negatively affect the fitness of some insect hosts, such as *S. frugiperda* [38]. If antibiotics affected larval development through toxicity or antifeedant effects, a greater or equal negative impact would be expected when administered in the 5th instar rather than in the 4th instar. Reintroducing a community of gut bacteria dissected from field–collected *C. medinalis* to 5th instar larvae that had been treated with antibiotics recovered some of the fitness traits, including adult fecundity. Currently, the degree to which gut microbes contribute to host development and fitness in Lepidoptera is debated [39,40,41,42]. Only limited studies showed significant effects of gut bacterial communities on caterpillar–plant interactions [15,26,31,41,43,44]. Mason et al. [15] reported interactions between plant defenses and gut bacteria, such that plant defenses that compromised the integrity of gut barriers in *S. frugiperda* larvae allowed some commensal bacteria to become opportunistic pathogens. Thus, the effects of gut bacterial community may be context–dependent, such as when the host is under dietary stress [45]. The role of bacterial communities in shaping insects’ fitness will, to some extent, vary depending on insect species and collection sites and sources.

Lastly, the composition of lepidopteran gut bacterial communities varies with geographic location and host plants [46,47,48]. In our study, we collected our field samples from rice paddies but applied the bacterial isolates and communities onto wheat leaves. Some individual isolates had detrimental effects on *C. medinalis* survival, which might differ if applied to rice leaves. Since the inoculation of the bacterial community from rice paddy larvae into wheat–reared antibiotic–treated larvae still exhibited beneficial effects, there may be mutualistic bacteria in *C. medinalis* that are present across a variety of host plants. Some individual isolates had detrimental effects on *C. medinalis* survival, which might vary if applied to rice leaves. Further research is needed to untangle the interactions between gut bacteria and host plants, as well as complex interactions among gut bacteria, which may vary on different host plants.

## 5. Conclusions

Our investigation into the effects of gut bacteria on *C. medinalis* fitness offers promising insights for developing novel and sustainable pest management strategies. We isolated 15 gut bacterial isolates using a culture–dependent method and identified with 16S rRNA sequencing. Alternative methods, such as multiple–locus variable number of tandem repeat analysis (MLVA), multilocus sequence analysis (MLSA), multilocus sequence typing (MLST), whole genome sequencing (WGS), and phenotypic–based techniques, could be applied to achieve more accurate bacterial species identification. Importantly, four of these isolates exerted strong adverse effects on female fecundity and 10 isolates caused high mortality. We also identified the importance of the bacterial community on improving development and fitness of the host. The knowledge gained from this study will be used in future studies of multitrophic interactions among microbes, plants, and *C. medinalis* to understand how the physical and chemical characteristics of crop varieties influence the interactions between the hosts and their microbes. This could lead to the selection of crop characteristics that leverage the deleterious effects of some naturally occurring gut bacteria on *C. medinalis* fitness.

## Figures and Tables

**Figure 1 insects-15-00947-f001:**
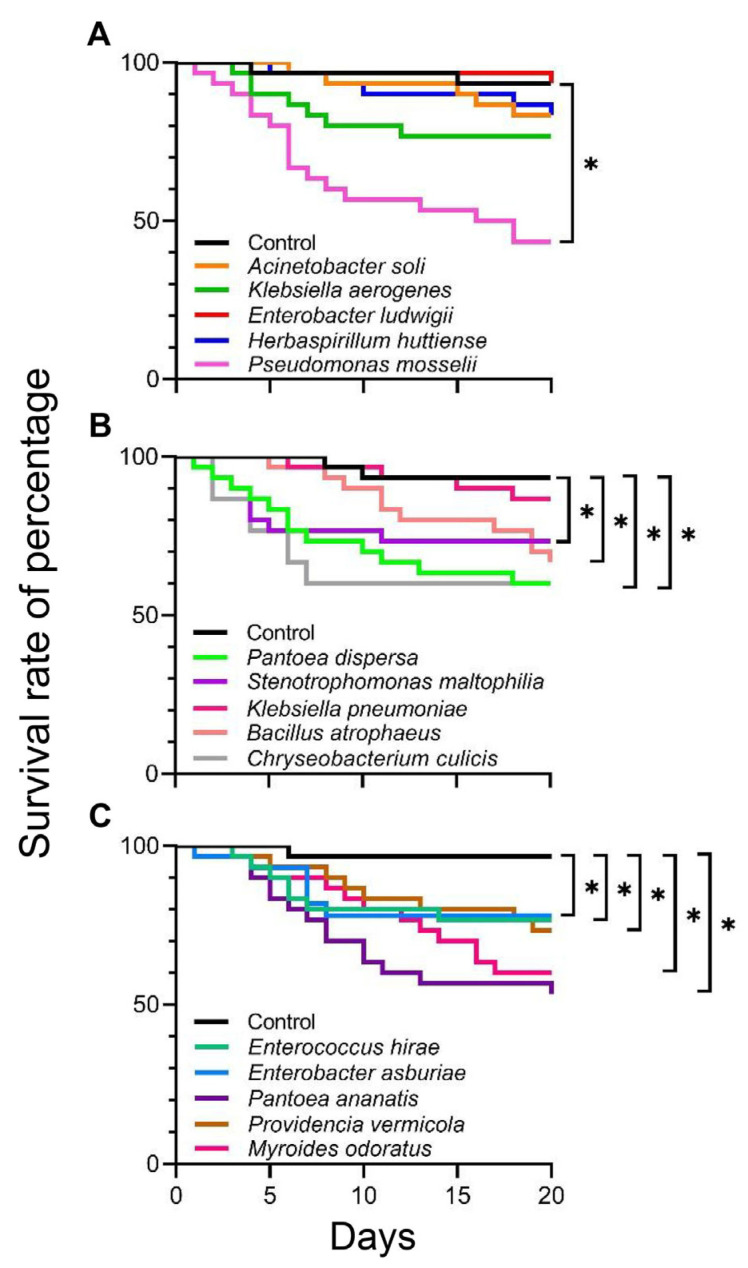
Kaplan–Meier survivorship estimates of *Cnaphalocrocis medinalis* larvae fed from the neonate to pupation stage with leaves treated with single bacterial isolates, which were obtained from field–collected larvae. Asterisk indicates a statistically significant difference compared with the controls at *α* < 0.05.

**Figure 2 insects-15-00947-f002:**
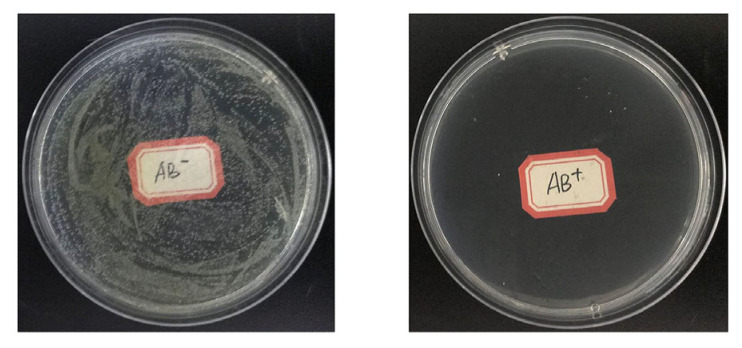
Growth of bacteria from homogenized guts of *Cnaphalocrocis medinalis* larvae that had been fed leaves treated with an antibiotic cocktail (AB^+^) or not treated with antibiotics (AB^–^).

**Figure 3 insects-15-00947-f003:**
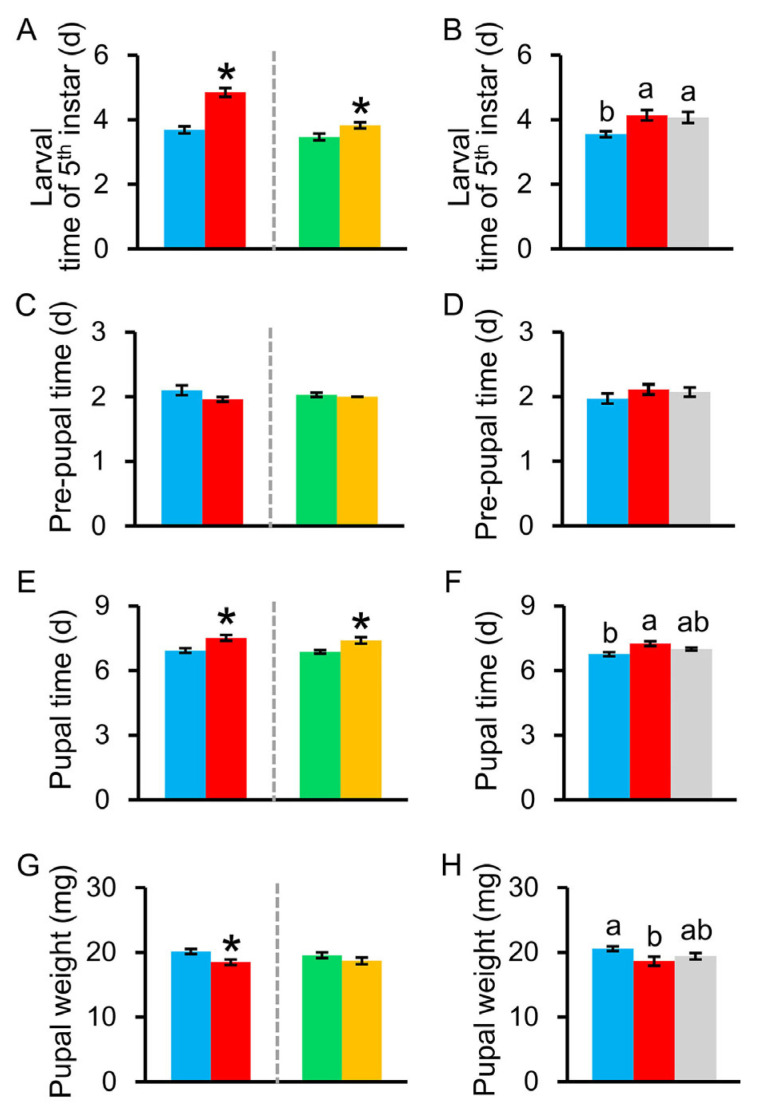
Effects of larval gut bacteria on fitness measures of *Cnaphalocrocis medinalis* during the immature stage, including (**A**,**B**) the larval development time of the 5th instar, (**C**,**D**) pre–pupal time, (**E**,**F**) pupal time, and (**G**,**H**) pupal weight. Red bar represents *C. medinalis* larvae that were fed leaves treated with an antibiotic cocktail for the duration of the 4th instar (AB^+^) and the blue bar represents larvae that were not given any antibiotics (AB^−^). Orange bar represents *C. medinalis* larvae that were fed leaves treated with an antibiotic cocktail for the duration of the 5th instar (AB^+^) and the green bar represents larvae that were not given any antibiotics (AB^−^). Gray bar represents *C. medinalis* larvae that were inoculated with a bacterial community in the 5th instar after being fed antibiotics in the 4th instar. Values are means ± SE. Asterisk indicates a statistically significant difference compared with the controls at *p* < 0.05. Different letters represent significant differences between treatments at *p* < 0.05.

**Figure 4 insects-15-00947-f004:**
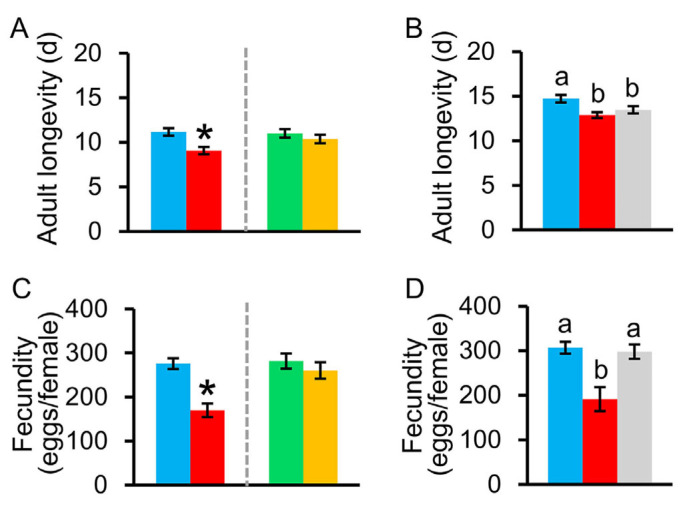
Effects of larval gut bacteria on the fitness measures of *Cnaphalocrocis medinalis* during the adult stage, including (**A**,**B**) adult longevity and (**C**,**D**) fecundity (i.e., total eggs laid per female). Red bar represents *C. medinalis* larvae that were fed leaves treated with an antibiotic cocktail for the duration of the 4th instar (AB^+^) and the blue bar represents larvae that were not given any antibiotics (AB^−^). Orange bar represents *C. medinalis* larvae that were fed leaves treated with an antibiotic cocktail for the duration of the 5th instar (AB^+^) and the green bar represents larvae that were not given any antibiotics (AB^−^). Gray bar represents *C. medinalis* larvae that were inoculated with a bacterial community in the 5th instar after being fed antibiotics in the 4th instar. Values are means ± SE. Asterisk indicates a statistically significant difference compared with the controls at *p* < 0.05. Different letters represent significant differences between treatments at *p* < 0.05.

**Table 1 insects-15-00947-t001:** Bacterial isolates identified from the guts of field–collected *Cnaphalocrocis medinalis* larvae and classified with the NCBI (National Center for Biotechnology Information) database.

Sample	NCBI Closest Match	Identity	Identification	GenBank No.
1	MT081629.1	99.88%	*Acinetobacter soli*	PP955231
2	MN173944.1	99.52%	*Klebsiella aerogenes*	PP955232
3	KF938661.1	99.76%	*Enterobacter ludwigii*	PP955233
4	MK064226.1	99.64%	*Herbaspirillum huttiense*	PP955234
5	MT598025.1	99.88%	*Pseudomonas mosselii*	PP955235
6	KF542916.1	99.79%	*Pantoea dispersa*	PP955236
7	MG905289.1	99.93%	*Stenotrophomonas maltophilia*	PP955237
8	MT081681.1	99.76%	*Klebsiella pneumoniae*	PP955238
9	KX225450.1	99.76%	*Bacillus atrophaeus*	PP955239
10	MN889357.1	99.76%	*Chryseobacterium culicis*	PP955240
11	KF040095.1	99.52%	*Enterococcus hirae*	PP955241
12	MK641844.1	99.88%	*Enterobacter asburiae*	PP955242
13	KX161909.1	99.76%	*Pantoea ananatis*	PP955243
14	KU870748.1	99.58%	*Providencia vermicola*	PP955244
15	MN833504.1	99.57%	*Myroides odoratus*	PP955245

**Table 2 insects-15-00947-t002:** Annotations at each classification level of bacterial isolates identified from the gut of field–collected *Cnaphalocrocis medinalis* larvae using the RDP (Ribosomal Database Project) database.

Isolates	Phylum	Class	Order	Family	Genus
*Acinetobacter soli*	Proteobacteria	Gammproteobacteria	Pseudomonadales	Moraxellacea	Acinetobacter
*Klebsiella aerogenes*	Proteobacteria	Gammproteobacteria	Enterobacteriales	Enterobacteriaceae	Unclassified
*Enterobacter ludwigii*	Proteobacteria	Gammproteobacteria	Enterobacteriales	Enterobacteriaceae	Enterobacter
*Herbaspirillum huttiense*	Proteobacteria	Betaproteobacteria	Burkholderiales	Oxalobacteraceae	Herbaspirillum
*Pseudomonas mosselii*	Proteobacteria	Gammproteobacteria	Pseudomonadales	Pseudomonadaceae	Pseudomonas
*Pantoea dispersa*	Proteobacteria	Gammproteobacteria	Enterobacteriales	Erwiniaceae	Pantoea
*Stenotrophomonas maltophilia*	Proteobacteria	Gammproteobacteria	Enterobacteriales	Erwiniaceae	Pantoea
*Klebsiella pneumoniae*	Proteobacteria	Gammproteobacteria	Enterobacteriales	Enterobacteriaceae	Klebsiella
*Bacillus atrophaeus*	Firmicutes	Bacilli	Bacillales	Bacillaceae	Bacillus
*Chryseobacterium culicis*	Bacteroidetes	Flavobacteriaceae	Flavobacteriales	Flavobacteriaceae	Chryseobacterium
*Enterococcus hirae*	Firmicutes	Bacilli	Lactobacillale	Enterococcaceae	Enterococcus
*Enterobacter asburiae*	Firmicutes	Bacilli	Lactobacillales	Enterococcaceae	Enterococcus
*Pantoea ananatis*	Proteobacteria	Gammproteobacteria	Enterobacteriales	Enterobacteriaceae	Pantoea
*Providencia vermicola*	Proteobacteria	Gammproteobacteria	Enterobacteriales	Morganellaceae	Providencia
*Myroides odoratus*	Bacteroidetes	Flavobacteriaceae	Flavobacteriales	Flavobacteriaceae	Myroides

**Table 3 insects-15-00947-t003:** Development time of *Cnaphalocrocis medinalis* larvae treated with different isolates of gut bacteria. Values are means ± SE. Asterisk indicates a statistically significant difference compared with the controls at *p* < 0.05.

Groups	Treatments	1st Instar Days ± SE	2nd Instar Days ± SE	3rd Instar Days ± SE	4th Instar Days ± SE	5th Instar Days ± SE	Total Larva Days ± SE
1	Control	3.68 ± 0.10	2.14 ± 0.10	2.07 ± 0.07	2.21 ± 0.12	3.86 ± 0.15	14.00 ± 0.19
	*Acinetobacter soli*	3.56 ± 0.12	2.16 ± 0.08	2.12 ± 0.07	2.24 ± 0.12	3.64 ± 0.14	13.72 ± 0.20
	*Klebsiella aerogenes*	3.48 ± 0.15	2.35 ± 0.10	2.22 ± 0.14	2.70 ± 0.12 *	3.70 ± 0.15	14.39 ± 0.29
	*Enterobacter ludwigii*	3.82 ± 0.09	2.25 ± 0.08	2.32 ± 0.09	2.36 ± 0.12	3.82 ± 0.13	14.57 ± 0.17 *
	*Herbaspirillum huttiense*	3.96 ± 0.11	2.20 ± 0.08	2.16 ± 0.11	2.44 ± 0.13	3.96 ± 0.17	14.72 ± 0.21 *
	*Pseudomonas mosselii*	4.00 ± 0.16	2.46 ± 0.18	2.92 ± 0.14 *	2.23 ± 0.20	4.38 ± 0.14 *	15.77 ± 0.46 *
2	Control	3.82 ± 0.15	2.50 ± 0.13	2.29 ± 0.10	2.39 ± 0.14	3.46 ± 0.16	14.46 ± 0.23
	*Pantoea dispersa*	3.50 ± 0.20	2.28 ± 0.11	2.56 ± 0.17	2.72 ± 0.18	4.56 ± 0.19 *	15.61 ± 0.38
	*Stenotrophomonas maltophilia*	3.95 ± 0.14	2.64 ± 0.16	2.36 ± 0.14	3.05 ± 0.14 *	4.05 ± 0.14 *	16.10 ± 0.27 *
	*Klebsiella pneumoniae*	3.77 ± 0.14	2.42 ± 0.10	2.38 ± 0.10	2.65 ± 0.16	3.73 ± 0.14	15.00 ± 0.30
	*Bacillus atrophaeus*	4.14 ± 0.22	2.52 ± 0.11	2.48 ± 0.15	2.62 ± 0.13	3.86 ± 0.14	15.62 ± 0.38 *
	*Chryseobacterium culicis*	3.76 ± 0.16	2.71 ± 0.17	2.18 ± 0.10	2.65 ± 0.17	3.71 ± 0.14	15.00 ± 0.35
3	Control	3.34 ± 0.11	2.24 ± 0.10	2.17 ± 0.09	2.48 ± 0.15	3.90 ± 0.10	14.14 ± 0.17
	*Enterococcus hirae*	4.35 ± 0.12 *	2.57 ± 0.11 *	2.61 ± 0.18	2.74 ± 0.17	4.26 ± 0.13 *	16.52 ± 0.36 *
	*Enterobacter asburiae*	3.79 ± 0.18 *	2.47 ± 0.12	2.26 ± 0.17	2.58 ± 0.16	3.89 ± 0.15	15.00 ± 0.43
	*Pantoea ananatis*	3.94 ± 0.25 *	2.63 ± 0.18	2.44 ± 0.16	2.44 ± 0.13 *	4.44 ± 0.18	15.88 ± 0.41 *
	*Providencia vermicola*	3.27 ± 0.12	2.59 ± 0.18	2.91 ± 0.19 *	3.18 ± 0.17 *	4.09 ± 0.16	16.05 ± 0.37 *
	*Myroides odoratus*	3.72 ± 0.16	2.28 ± 0.14	2.44 ± 0.12	2.44 ± 0.15	4.28 ± 0.14 *	15.17 ± 0.29 *

**Table 4 insects-15-00947-t004:** Development time of pre–pupa and pupa, and pupal weight, female ratio, adult longevity, and fecundity of *Cnaphalocrocis medinalis* larvae treated with different gut bacteria. Values are means ± SE. Asterisk indicates a statistically significant difference compared with the controls at *p* < 0.05.

Groups	Treatments	Pre–Pupa Days ± SE	Pupa Days ± SE	Pupal Weight mg ± SE	Female Ratio	Adult Longevity Days ± SE	Fecundity Eggs/Female
1	Control	2.11 ± 0.08	7.07 ± 0.13	21.21 ± 0.46	0.43	13.18 ± 0.51	305.33 ± 22.90
	*Acinetobacter soli*	2.08 ± 0.06	6.88 ± 0.12	20.84 ± 0.47	0.48	10.76 ± 0.50 *	194.57 ± 23.58 *
	*Klebsiella aerogenes*	2.04 ± 0.04	7.04 ± 0.12	22.27 ± 0.45	0.48	11.39 ± 0.66 *	354.25 ± 32.42
	*Enterobacter ludwigii*	2.21 ± 0.15	7.07 ± 0.17	21.65 ± 0.48	0.46	12.21 ± 0.35	348.18 ± 16.80
	*Herbaspirillum huttiense*	2.12 ± 0.07	6.96 ± 0.11	21.91 ± 0.42	0.40	10.84 ± 0.57 *	300.67 ± 12.48
	*Pseudomonas mosselii*	2.00 ± 0.16	7.15 ± 0.27	21.88 ± 0.88	0.54	12.31 ± 0.38	318.50 ± 45.21
2	Control	2.00 ± 0.05	7.14 ± 0.15	21.85 ± 0.69	0.50	12.64 ± 0.46	316.79 ± 21.00
	*Pantoea dispersa*	2.11 ± 0.08	7.11 ± 0.08	17.42 ± 0.60 *	0.50	10.39 ± 0.63 *	281.50 ± 16.43
	*Stenotrophomonas maltophilia*	2.05 ± 0.08	7.41 ± 0.13	19.79 ± 0.44 *	0.50	10.64 ± 0.47 *	208.38 ± 38.54 *
	*Klebsiella pneumoniae*	2.04 ± 0.04	7.50 ± 0.28	18.71 ± 0.63 *	0.50	11.50 ± 0.63	321.73 ± 17.34
	*Bacillus atrophaeus*	2.00 ± 0.00	7.33 ± 0.13	18.73 ± 0.49 *	0.62	11.76 ± 0.64	203.91 ± 30.87 *
	*Chryseobacterium culicis*	2.06 ± 0.10	7.29 ± 0.17	19.05 ± 0.73 *	0.47	12.76 ± 0.60	277.43 ± 35.85
3	Control	2.03 ± 0.06	6.86 ± 0.08	19.74 ± 0.42	0.48	11.97 ± 0.46	281.00 ± 19.49
	*Enterococcus hirae*	2.09 ± 0.11	7.22 ± 0.11 *	21.48 ± 0.60 *	0.48	10.61 ± 0.63	279.64 ± 8.49
	*Enterobacter asburiae*	1.89 ± 0.07	7.21 ± 0.15 *	19.62 ± 0.49	0.47	10.68 ± 0.57	286.67 ± 24.00
	*Pantoea ananatis*	2.06 ± 0.06	7.44 ± 0.16 *	19.14 ± 0.37	0.44	10.81 ± 0.48	292.00 ± 18.23
	*Providencia vermicola*	2.14 ± 0.08	7.95 ± 0.31 *	19.05 ± 0.43	0.45	11.36 ± 0.54	210.89 ± 26.47 *
	*Myroides odoratus*	2.11 ± 0.08	6.89 ± 0.18	18.84 ± 0.520	0.50	12.17 ± 0.40	294.50 ± 19.64

## Data Availability

Please request any particular data through the corresponding author. Sequence data of the 15 bacterial isolates were submitted to the NCBI database (accession number PP955231–PP955245).

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
