# Peer review of "Effects of Gut Bacteria on the Fitness of Rice Leaf Folder Cnaphalocrocis medinalis"

_insects, 2024, doi:10.3390/insects15120947_

Round 1

Reviewer 1 Report

Comments and Suggestions for Authors

I think the manuscript is interesting and that many researchers in the field may be interested to take a look on. The study contributes valuable insights into the role of gut microbiota in C. medinalis and their potential as biocontrol agents.

Introduction: 

-While the introduction provides a wealth of information, some sentences are overly lengthy and could benefit from simplification. For example, lines 83–88 describing the experimental approach are verbose and could be restructured for clarity.

-The transition between the general discussion of gut microbes and the specific case of C. medinalis is abrupt. The paragraph starting at line 72 could better bridge the broad context with the specific pest by more explicitly tying microbial roles to the agricultural importance of C. medinalis.

-While the limitations of NGS are mentioned (lines 69–71), the advantages of culture-dependent methods are not as thoroughly discussed. Expanding on how these methods uniquely contribute to functional assessments would strengthen the rationale.

-Certain points are repeated unnecessarily. The roles of gut bacteria in influencing host fitness and as potential biocontrol agents are stated multiple times, which could be streamlined to avoid redundancy.

-Some claims lack sufficient citation. For example, the introduction mentions the migratory propensity of C. medinalis adults (line 77) without directly citing supporting studies. Ensuring all key statements are backed by references enhances credibility.

-Although the study aims to fill gaps in understanding the role of gut bacteria in C. medinalis, the introduction does not articulate a specific hypothesis. Explicitly stating the novelty of the work compared to prior studies would strengthen its impact.

M&M:

-The centrifugation at 6,000 rpm for 30 minutes to remove debris could be critiqued. The relative centrifugal force (RCF) should be specified if the name of the centrifuge and rotor is not specified.

Results:

-The 16S rRNA analysis provides sufficient resolution for genus-level identification but is insufficient for accurate species-level classification. Therefore, the authors should clarify that the identification is putative, based solely on 16S rRNA analysis, and emphasize the need for complementary methods (Multi-Locus Sequence Analysis (MLSA), whole-genome sequencing, or phenotypic characterization) in order to achieve a more precise and reliable species-level identification, and this clarification should be included in the manuscript. Very important.

Discussion:

-The article would benefit from more detailed discussions of the mechanisms behind bacterial interactions with the host and more thorough exploration of practical applications.

-While the authors mention the need for further research, the discussion could be more specific in outlining future experimental directions.

Reviewer 2 Report

Comments and Suggestions for Authors

The manuscript offers a detailed study on the role of gut bacteria in influencing the fitness of Cnaphalocrocis medinalis. It presents significant findings that contribute to pest management and microbiology research. With minor revisions to improve clarity and structure, the paper could be further strengthened.

Here are some comments and suggestions based on the initial review:

Line 63, 65, add “the” before “effects” and “larvae”, respectively

Line 69, “yet research evidence of the functions of these gut microbes remains limited” could be rewritten as “yet research on the functions of these gut microbes remains limited”

Line 70-72, Please rephrase the sentence more concise and better readability and precision.

Line 103, “undergo” should be modified as “underwent”

Line 118, add “while” before “the remaining”

Table 1 should not appear in Materials and Methods

Line 122, 126, 141, Please provide the full name of YT.

Line 60-61, state not clearly. I think if the individuals died at the 5th instar, the development time of 1st to 4th instars could be counted.

Line179-180, It is vague. Should rephrase.

2.5 and 2.6 can be merged into one title, both studying the impact of gut microbiota on the fitness of rice leaf rollers.

Line 212-214, “The gene sequences…BLAST” should be moved to the part of “material and methods”

Line 220, This sentence is not precise and not all isolates have a significant impacts on survival.

Line 221-223, use survival rate, instead of mortality. The sentence could be divided into two sentence.

Line 226-227, rewrite this sentence.

The name of y-axis in fig. 1 should be survival rate of percentage, instead of probability. And the title, at α = 0.05, not at P < 0.05.

Tables 1 and 2 require additional sample sizes for each statistical value. Suggest using the the age-stage two sex life table (Chi, 1988) for analysis.

3.2, 3.3, 3.4, it is recommended to merge them into one title with several subheadings. It is recommended to separate the statistics of male and female insects, especially as the weight of pupae and the lifespan of adults are greatly affected by the proportion of females.

Figure 2 can be included in the supplementary materials to demonstrate the bactericidal effect of AB, but it cannot appear in the results of 3.5.

3.5 and 3.6, it is recommended to merge them under one heading

In single bacterium experiments, it is best to present the data of changes in the single bacterium abundance in the gut.

How to eliminate the negative impact of antibiotics on the fitness of rice leaf rollers seems to have not been taken into account in the research.

Lack of experiments on the replenishment of bacterial community suspension from laboratory colony into AB+ laboratory populations to verify the effect of antibiotics on fitness.

It would be best to provide data on the colonization of gut microbiota for replenishment in the gut.

Comments on the Quality of English Language

Some english expression needs polishing. 

Reviewer 3 Report

Comments and Suggestions for Authors

The paper adds a reliable set of data on bacterial community roles in a notorious pest of cereals, the rice leaf folder, Cnaphalocris medinalis. As noted in the Introduction, NGS-based studies have already shown variation of gut microbiota in this dangerous pest. The novelty of the present study is explained by isolation and cultivation of the predominate bacterial species and functional analysis of their effects. The experimental design is robust, the results are straightforward, the conclusions are convincing. Among approaches exploited by the authors, antibiotic treatment and re-inoculation of the bacterial community fraction can be considered as a sound and robust tool to reach the goals persuaded by the  researchers. Other researchers should follow this experimental pipeline aimed to further explore interactions with gut microbiota of this and other pest species. It is interesting if highly virulent bacterial strains showing ponetial for microbial pest control, can be found through these studies. I recommend publication after minor corrections of the text, mainly concerning methodological description and grammar, as proposed below.

18 and elsewhere: after first mentioning of the genus epithet, it should be contracted in combinations of [genus name][species name]

97-98 what is the year of insect collection?

100-101 how many adults per cup?

101-102 what kind of “plastic film”, how ventilation was performed?

116 buffers of each larval gut tissue – how many “buffers” per “each tissue”?

122 dilutions were spread – I doubt a “dilution” can be spread, consider rephrasing, find an appropriate subject

126-127 please give a very brief description of the method, in addition to the reference given, so that it can be perceived without reading Pan et al.

127 analyzed with NCBI – does it imply participation of the NCBI team in the analysis?

Table 1. Please add higher level taxonomy of each species

136 and elsewhere: a phrase shouldn’t start with a contracted genus epithet, give it in full

147 and elsewhere: Petri is a personal name, should start with a capital letter

152 the treated leaves … were replaced – what the treated leaves were replaced with? With newly treated leaves or with untreated leaves?

163-164 please provide the brand and the sensitivity of the micro-balance

197-169 no need to mention the producer three times, it’s enough to say “Three antibiotics (AB), all from Sangon Biotech”. By the way, what the numbers stay for?

175 vs 182 there were either 4th or 5th instar and 30 larvae per treatment. Does it mean separate groups of 30 4th instars and 30 5th instars, or a single group of mixed 4th and 5th instars? In the latter case ,what was the proportion of 4th:5th instars? How the possible bias of these two instars was excluded?

212-214 it mainly repeats information already given is Methods. The whole section should be extended. You may indicate which phyla were presented by this list of species.

212 It would be nice to explain why did you used RDP if it only resolves generic level

375 The variations among these isolates – variation of what?

378-386 you briefly summarize the effects of the studied isolates on the fecundity and larval instar duration, but do not compare these data to your observations on adult lifespan duration, described in the Results.

394-395 please give examples of similar studies where bacteria were isolated from guts and utilized for microbial control applications. Other examples showing the use of knowledge of bacterial communities of insect pests for practice are also welcome, considering your statements in Liens 434-435.

Comments on the Quality of English Language

The grammar and style and predominately fine. The text contains some grammar impurities (“research … have”, “effect were”, “longevity were” etc) and few typos (“filed-collected” etc). Style also requires slight polishing. For example, there is no need to use “developmental durations” and “instar durations” in multiple form. In Line 14, in the phrase “this pest possesses diverse bacterial communities in their gut” it is not clear whose  gut is implied by “their”, because the subject “pest” is singular. In Line 316, the collocation “lighter pupal weight” doesn’t’ seem to be correct, as being “lighter” is a property of pupae (“lighter pupae”) and not of weight. The weight can be higher or lower, increased or decreased, enlarged or reduced, but not “lighter”.
